# Bioinformatics Approach to Identifying Molecular Targets of Isoliquiritigenin Affecting Chronic Obstructive Pulmonary Disease: A Machine Learning Pharmacology Study

**DOI:** 10.3390/ijms26083907

**Published:** 2025-04-21

**Authors:** Sha Huang, Lulu Zhang, Xiaoju Liu

**Affiliations:** 1The First Clinical Medical College, Lanzhou University, Lanzhou 730000, China; shahuangelsa@163.com (S.H.);; 2Department of Gerontal Respiratory Medicine, The First Hospital of Lanzhou University, Lanzhou 730000, China

**Keywords:** alveolar macrophages, bioinformatics, glycolysis, phagocytosis, COPD, isoliquiritigenin

## Abstract

To identify the molecular targets and possible mechanisms of isoliquiritigenin (ISO) in affecting chronic obstructive pulmonary disease (COPD) by regulating the glycolysis and phagocytosis of alveolar macrophages (AM). Datasets GSE130928 and GSE13896 were downloaded from the Gene Expression Omnibus (GEO) database. Genes related to glycolysis and phagocytosis phenotypes were obtained from the GeneCards and MSigDB databases, respectively. Weighted gene co-expression network analysis (WGCNA) and differential analysis were conducted on GSE130928 to identify potential target genes for COPD (gene list 1). ISO target genes were gathered from the Traditional Chinese Medicine System Pharmacology (TCMSP) database, as well as the Comparative Toxicogenomic Database (CTD) and PubChem databases (gene list 2). COPD-related targets were gathered from the CTD and GeneCards databases, and the predicted targets of COPD were obtained by taking the intersection of these sources (gene list 3). From the three gene lists, key pathways were identified. The protein–protein interaction (PPI) network was created by extracting the common genes found in all key pathways and ISO targets. Candidate therapeutic targets were identified using the Minimum Common Oncology Data Element (MCODE) algorithm. These targets were then intersected with glycolysis and phagocytic phenotype-associated genes. The resulting intersection underwent further screening using eight distinct machine learning methods to identify phenotype-related key therapeutic targets. Clinical diagnostic evaluations—including nomogram analysis, receiver operating characteristic (ROC) analysis, correlation studies, and inter-group expression comparisons—were subsequently performed on these key targets. The research findings were validated using the independent dataset GSE13896. Additionally, gene set enrichment analysis (GSEA) was conducted to explore their functional relevance. Immune cell profiling was performed using mRNA expression data from AM in COPD patients. Molecular docking was then carried out to predict interactions between ISO and the identified key target genes. Differential expression analysis and WGCNA module analysis identified a total of 890 potential targets for COPD. Additionally, 3265 predicted targets for COPD were obtained through the intersection of two disease databases. Database searches also yielded 142 targets for ISO. Enrichment analysis identified 20 key pathways, from which three key targets (*AKT1*, *IFNG*, and *JUN*) were ultimately selected based on their overlap with enriched genes, ISO targets, and glycolysis- and phagocytosis-related genes. They were also validated using external datasets. Further analysis of signaling pathways and immune cell profiles highlighted the influence of immune infiltration in COPD and underscored the critical role of macrophages in disease pathology. Molecular docking simulations predicted the binding interactions between ISO and the three key targets. *AKT1*, *IFNG*, and *JUN* may be the key targets of ISO in regulating glycolysis and phagocytosis to affect COPD.

## 1. Introduction

In 2017, the number of people with chronic respiratory diseases was estimated to be 544 million, of which chronic obstructive pulmonary disease (COPD) accounted for 55%. The prevalence of COPD is likely to increase due to the aging population [1]. COPD is the third leading cause of death worldwide [1]. Studies have shown that 24% of COPD patients die within 5 years of diagnosis [2]. COPD costs the U.S. healthcare system $32 billion per year in direct costs [3]. A Danish cohort study showed that total direct costs for COPD patients were three times higher than those for controls, and costs increased with disease severity [4]. This shows that the disease and economic burden of COPD is heavy, so there are constant new explorations in the drug treatment of COPD.

Recent studies have shown that the alveolar macrophages (AM) of COPD patients overrely on glycolysis as a source of ATP, resulting in a decrease in energy status and an inability to fully maintain macrophage phagocytosis and other functions [5]. Modulating glycolysis and phagocytosis has been shown to be effective in patients with COPD [6,7,8].

Isoliquiritigenin (ISO) is a flavonoid compound and one of the most biologically active chalcones derived from the roots of licorice and other plants [9]. ISO demonstrates therapeutic potential against gastric cancer by decreasing GLUT4-mediated glucose uptake and inhibiting the protein expression of c-Myc and HIF-1α, ultimately inducing a collapse of energy metabolism [10]. In colorectal cancer, ISO exerts its effects by inhibiting glycolysis and lactate production through the modulation of AMPK and the Akt/mTOR signaling pathways [11]. Furthermore, in non-tumor diseases, ISO enhances the phagocytic activity of immune cells by regulating SLC7A11-mediated glycolysis, thereby facilitating wound healing in diabetic conditions [12]. ISO also enhances the phagocytosis of RAW264.7 treated with a toxic substance (BDE-47) and alleviates the immune function damage caused by BDE-47 [13]. Furthermore, ISO has demonstrated protective effects against cigarette smoke-induced COPD in mouse models [14].

We speculate that ISO may also affect COPD by affecting the glycolysis and phagocytic function of COPD, but the possible target of action is still unclear. Therefore, this study aims to investigate the molecular targets through which ISO influences COPD by regulating glycolysis and phagocytosis. To achieve this, we employed bioinformatics approaches, including differential expression analysis, weighted gene co-expression network analysis (WGCNA), Kyoto Encyclopedia of Genes and Genomes (KEGG) pathway analysis, and machine learning. These methods were integrated with analyses of signaling pathways and immune cell profiles to explore the mechanisms and molecular targets by which ISO modulates glycolysis and phagocytosis in the context of COPD.

## 2. Results

### 2.1. Identification of COPD-Related Potential Target Genes

To identify genes involved in COPD pathophysiology, AM mRNA chip data were normalized and subjected to differential expression analysis. By applying the screening criteria of |logFC| > 0.25 and adjP < 0.05, we identified a total of 2280 differentially expressed genes (DEGs) in the GSE130928 microarray dataset, comprising 1108 upregulated genes and 1172 downregulated genes. The results of the normalization and differential expression analysis are visualized in Figure 1A and Figure 1B, while the volcano plot and heatmap of the DEGs are shown in Figure 1C and Figure 1D, respectively.

To explore the relationships between genes and identify co-expression modules relevant to COPD, the expression values of the top 5414 DEGs were analyzed using WGCNA. Samples were clustered based on the Pearson correlation coefficient, and a sample clustering tree was generated to assess data quality. The optimal soft threshold (β = 9) was chosen based on the scale-free topology standard (R^2^ = 0.85) to construct a scale-free co-expression network (Figure 2A,B). Using the dynamic tree-cutting method and a dissimilarity threshold (DissThres) of 0.25, a total of 18 modules were obtained after merging similar modules. Each module was assigned a unique color: red (318 genes), blue (447 genes), magenta (285 genes), lightgreen (119 genes), greenyellow (243 genes), or lightcyan (162 genes). The module assignment results are visualized in Figure 2C (24 cases in the normal group and 22 cases in the COPD group). The correlation between modules and COPD was evaluated by calculating the MM and GS, and the results are presented as heatmaps in Figure 2D–I.

To further refine the analysis, genes from the red, blue, magenta, light green, greenyellow, and lightcyan modules were intersected with the DEGs identified earlier. This intersection yielded 890 potential therapeutic targets associated with COPD (Figure 3A). A total of 142 ISO targets were identified from the database. Using “chronic obstructive pulmonary disease” as a keyword, predicted target genes associated with COPD were retrieved from the Comparative Toxicogenomic Database (CTD) and GeneCards databases, resulting in 3265 potential COPD-related targets (Figure 3C). KEGG pathway enrichment analysis was subsequently conducted on three sets of genes: (1) the overlapping genes between DEGs and key WGCNA modules, (2) the predicted COPD targets derived from the two databases, and (3) the ISO target genes. These analyses produced three enriched pathway sets: KEGGa, KEGGb, and KEGGc (Figure 3B,D,E). Finally, the three pathway sets were intersected to identify 20 key pathways. The number of genes enriched in each pathway and their corresponding *p*-values are detailed in Table 1.

### 2.2. Construct Drug Small Molecule–Target–Pathway Network and Protein–Protein Interaction (PPI) Network and Screen Key Networks

All genes enriched in the 20 key pathways were combined with the ISO target genes, resulting in 54 predicted therapeutic target genes (Figure 4A). As outlined in the methodology, the predicted molecular targets of ISO were sourced from the Traditional Chinese Medicine System Pharmacology (TCMSP) database, as well as the CTD and PubChem databases. The “drug small molecule–target–pathway” network (Figure 4B) and the PPI network (Figure 4C) were subsequently generated. Using the MCODE (Minimum Common Oncology Data Element) plug-in for network analysis, a key subnetwork was identified, consisting of 40 candidate key genes (Figure 4D). These genes were further intersected with glycolysis and phagocytosis phenotypes, ultimately yielding 19 phenotype-related candidate genes: *AKT1*, *BCL2*, *CDKN1A*, *FOS*, *HSP90AA1*, *IFNG*, *IL6*, *JAK2*, *JUN*, *MAPK1*, *MAPK14*, *MAPK3*, *NFKB1*, *NOS2*, *PPARG*, *RELA*, *STAT3*, *TGFB1*, and *TNF*.

### 2.3. Machine Learning-Based Algorithms Identify Key Therapeutic Targets in COPD Patients

To further identify key therapeutic targets related to the COPD phenotype, machine learning algorithms were applied. The results from eight machine learning models (Figure 5A and Appendix A) were analyzed for intersections, and three key targets—*AKT1*, *IFNG*, and *JUN*—were identified as the most relevant for ISO’s therapeutic effects. These targets were selected for further analysis, and their diagnostic efficacy in COPD was evaluated (Figure 5B). To assess the clinical applicability of the findings, a COPD risk assessment nomogram was constructed using the Rms package in R (Figure 5C). We also validated this finding using the validation dataset GSE13896 (Appendix A).

### 2.4. Correlation Analysis of Key Targets, COPD Expression Profiles, and Gene Set Enrichment Analysis

To investigate the relationships among the three key targets, we conducted a correlation analysis (Figure 6A). The results revealed a negative correlation between *AKT1* and *JUN*, with no significant correlation observed for *IFNG*. We further examined the expression levels of *AKT1*, *IFNG*, and *JUN* in COPD patients compared to a normal control group (Figure 6B). Next, we performed gene set enrichment analysis (GSEA) to explore the functional roles of these three key targets in COPD (Figure 6C–E).

### 2.5. Immune Cell-Level Analysis

To investigate immune differences between the normal and COPD groups, we utilized the robust CIBERSORT deconvolution algorithm to analyze the composition of immune cells in the samples (Figure 7A). This sophisticated computational approach allowed us to estimate the proportions of 22 distinct immune cell types with high accuracy. Comparative analysis revealed significant differences in the distribution of specific immune cell populations between the two groups (Figure 7B). In the COPD group, levels of memory B cells, regulatory T cells (Tregs), gamma delta T cells (γδ T cells), M0 macrophages, and resting dendritic cells were elevated compared to the normal group. Conversely, levels of CD8+ T cells, resting memory CD4+ T cells, activated NK cells, M1 macrophages, activated mast cells, and eosinophils were reduced in the COPD group.

To better understand the functional relevance of these immune cell differences, we performed a correlation analysis to identify potential cell-to-cell relationships and their contributions to COPD pathology (Figure 7C). Given the critical role of AM in the onset and progression of COPD, we placed particular focus on macrophage-related interactions. M0 macrophages were significantly positively correlated with memory B cells, Tregs, γδ T cells, resting mast cells, and resting dendritic cells, while showing significant negative correlations with plasma cells, naïve CD4+ T cells, resting memory CD4+ T cells, activated NK cells, eosinophils, activated mast cells, activated dendritic cells, and M1 macrophages. M1 macrophages were significantly positively correlated with activated NK cells, neutrophils, eosinophils, and M2 macrophages but showed significant negative correlations with activated dendritic cells. M2 macrophages exhibited significant positive correlations with follicular helper T cells, Tregs, γδ T cells, neutrophils, and resting mast cells and significant negative correlations with naïve CD4+ T cells, resting memory CD4+ T cells, activated mast cells, and activated dendritic cells. Furthermore, we analyzed the correlations between each key target and the immune cell populations (Figure 7D).

### 2.6. Molecular Docking Validation

To evaluate the potential interactions between ISO and the three key targets in COPD, molecular docking simulations were performed with *AKT1*, *IFNG*, and *JUN*, respectively (Figure 8A–F, Table 2). The binding affinities of ISO with these targets ranged from −7.4 to −6.1 kcal/mol, all below the threshold of −5 kcal/mol. These results suggest that ISO exhibits strong binding potential to the key targets implicated in COPD (Figure 8B,D,F).

## 3. Discussion

Bioinformatics analyses and databases are widely utilized to enhance our understanding of human diseases and support advancements in disease diagnosis and prognosis. In this study, we leveraged KEGG pathway analyses to identify potential biomarkers associated with ISO, COPD, glycolysis, and phagocytosis. Using machine learning approaches, we identified *AKT1*, *IFNG*, and *JUN* as key targets through which ISO may regulate glycolysis and phagocytosis, ultimately influencing COPD progression. Key pathways (referred to as KEGGa, KEGGb, and KEGGc) were derived from datasets and multiple databases, and the results were cross-validated to minimize the likelihood of false positives. This study presents a novel approach to exploring COPD through the lens of pathways and immune infiltration. By integrating methods such as WGCNA, PPI network construction, machine learning, and molecular docking, we have elucidated the potential mechanisms by which ISO modulates glycolysis and phagocytosis to impact COPD development.

*AKT1*, a member of the *AKT* protein family, plays a critical role in cellular metabolism. Previous studies have shown that *AKT* can activate mTOR signaling and promote the expression of HIF-1α, which subsequently regulates glycolysis-related proteins such as GLUT and HK2. This process enhances glucose uptake and lactate production in endothelial cells [15]. Under hypoxic conditions, AKT has been reported to modulate PFKFB3, thereby regulating glycolysis in human monocyte-derived macrophages [16]. Additionally, AKT has been implicated in mediating enhanced phagocytic activity in these macrophages [17]. These findings align with the results of our study, suggesting that *AKT1* may serve as a key target for modulating glycolysis and phagocytic function in COPD.

After stimulation with *IFNG*, the metabolic state of bone marrow-derived macrophages gradually changes from oxidative phosphorylation to glycolysis; that is, *IFNG* can induce macrophage glycolysis conversion [18]. In other studies, *IFNG* inhibited phagocytosis in macrophages derived from the spleen, peritoneum, and the human monocytic leukemia cell line THP-1 [19,20,21]. Therefore, these findings support the result of this study that *IFNG* may be one of the important targets for regulating glycolysis and phagocytosis in COPD.

*JUN* interacts with NR3C2 to modulate glycolysis in pancreatic cancer [22]. Bioinformatics analysis also showed that *JUN* may bind to the GLUT1 promoter in breast cancer tissues, downregulating its gene expression or mRNA stability, thereby inhibiting glycolysis [23]. *JUN* promotes enhanced phagocytosis of THP-1-derived macrophages via CXCL14/ERK1/2 [24]. *JUN* activation enhances the phagocytosis of colorectal cancer cells by monocyte-differentiated macrophages [25]. Therefore, these findings support the result of this study that *JUN* may be one of the important targets for regulating glycolysis and phagocytosis in COPD.

This study suggests that ISO may be beneficial in the treatment of COPD. These findings align with those of Yu et al., who demonstrated that ISO mitigated lung pathological damage in a COPD mouse [14]. Furthermore, they reported that ISO exerts protective effects in COPD mice by modulating inflammation and oxidative stress, specifically through the modulation of myeloperoxidase activity, malondialdehyde levels, and the expression and activity of key signaling pathways, including nuclear factor erythroid 2-related factor 2 (Nrf2) and nuclear factor κB (NF-κB) [14]. ISO has also demonstrated therapeutic effects in other respiratory conditions, including acute lung injury and lung cancer [26,27].

The results of this study revealed that in COPD, the levels of immune cells, such as memory B cells and Tregs, were higher compared to the normal group, while the levels of immune cells, such as CD8+ T cells and M1 macrophages, were lower. These findings differ somewhat from those reported by Zhang et al. [28]. The discrepancy may be attributed to variations in tissue sources between the studies [29]. Our study focused on AM, whereas Zhang et al. analyzed lung tissue. The increased presence of immune cells, such as memory B cells and γδ T cells, may contribute to the initiation and amplification of inflammation, whereas the decreased levels of CD8+ T cells, M1 macrophages, and other immune cells could impair immune defense mechanisms and the clearance of inflammation.

The positive and negative correlations between macrophage subtypes and various immune cells highlight the dynamic changes within the complex immune microenvironment of the disease. M0 macrophages, which are unpolarized precursors, show a positive correlation with memory B cells, suggesting a potential joint role in enhancing immune responses or forming immune memory. This association might support processes like antibody production against pathogens or self-antigens. Additionally, their positive correlation with γδ T cells suggests involvement in the initiation or regulation of inflammatory responses. M0 macrophages are also positively correlated with resting mast cells, resting dendritic cells, and Tregs, indicating a possible role in maintaining immune homeostasis and curbing excessive inflammation in chronic inflammatory environments.

Conversely, the negative correlation between M0 and both naïve and resting memory CD4+ T cells suggests that M0 may inhibit the activation of naïve CD4+ T cells or the maintenance of memory CD4+ T cells. This inhibition could impair CD4+ T cell function, affecting cellular immune responses, reducing the body’s defense against pathogens, and potentially impacting the formation and maintenance of immune memory. The negative correlation with eosinophils indicates a regulatory relationship whereby M0 may suppress eosinophil activity. Furthermore, the negative correlation with plasma cells and activated immune cells, including activated NK cells, mast cells, and dendritic cells, suggests that M0 may limit acute inflammatory responses by inhibiting B cell differentiation and immune activation. This aligns with the “low-level continuous activation” state observed in chronic inflammation in COPD. Lastly, the negative correlation with M1 macrophages underscores the functional opposition between M0 and M1, suggesting that M0 may not participate in the pro-inflammatory phase.

M1 is a pro-inflammatory macrophage. Its positive correlation with neutrophils, eosinophils, and activated NK cells suggests that M1 may drive the acute inflammatory response of COPD, recruit and activate these cells, and lead to tissue damage. The positive correlation with M2 may indicate that there is a dynamic balance of macrophage polarization. M1 and M2 may coexist at different stages of COPD inflammation. For example, M1 dominates the acute inflammatory stage, while M2 plays a very important role in anti-inflammatory, repair, and remodeling processes [29,30]. The negative correlation with activated dendritic cells suggests that M1 macrophages may inhibit activated dendritic cells, affecting the antigen presentation function of dendritic cells and the activation of T cells.

M2 macrophages, known for their anti-inflammatory and tissue-repairing functions, demonstrate significant positive correlations with follicular helper T cells, Tregs, γδ T cells, neutrophils, and resting mast cells. These findings suggest that M2 macrophages may play a key role in resolving inflammation and promoting tissue repair. However, the observed negative correlations with naïve CD4+ T cells and resting memory CD4+ T cells imply that M2 macrophages may inhibit the activation of naïve CD4+ T cells and impair the function of memory CD4+ T cells. This inhibition could disrupt normal cellular immune responses, potentially hindering effective pathogen clearance and the formation of immune memory.

Additionally, the negative correlation between M2 macrophages and activated mast cells suggests that M2 macrophages may regulate the release of inflammatory mediators from mast cells, thereby controlling the intensity and duration of the inflammatory response. Similarly, the negative correlation with activated dendritic cells indicates a potential mutual inhibition between M2 macrophages and activated dendritic cells.

These findings highlight the importance of modulating the interactions between different immune cell types as a potential strategy for developing novel COPD treatments, which could lead to improved clinical outcomes [28,31]. Bioinformatics analysis provides valuable insights into the degree of immune cell infiltration, offering potential benefits for the early diagnosis and treatment of COPD.

This study has several limitations. First, it is based on publicly available datasets and database information, utilizing computational methods that have not undergone experimental validation (e.g., human, animal, in vitro, etc.). However, we have corroborated our findings using external datasets. Future research will aim to confirm these results through in vitro and/or in vivo experiments. Second, due to the specific focus on AM in COPD, there were limited options for public datasets. While we utilized GSE130928 and GSE13896 as training and validation sets to enhance the accuracy of our findings, the sample sizes analyzed were relatively small and may exhibit heterogeneity. Therefore, additional sequencing data with a larger sample size or further experimental validation are necessary to support the findings of our study. Finally, the public databases included in this study lack critical clinical information, including lung function, disease severity classification, comorbidities, medication use, and dietary intake. This absence hinders the ability to correlate the identified molecules with clinical data, and it also prevents adjustment for potential confounding factors that may influence the results.

## 4. Materials and Methods

### 4.1. Overall Study Design

In this study, we utilized several bioinformatics approaches to identify key therapeutic targets for COPD. Differential expression analysis and WGCNA were performed on an mRNA expression dataset of AM from COPD patients to identify DEGs and significant module genes (Appendix A). COPD-associated genes were retrieved from the CTD [32] and GeneCards [33] databases, and the intersection of these genes was used to obtain predicted COPD targets (Appendix A). Drug-related targets were obtained from the TCMSP [34], CTD [32], and PubChem [35] databases (Appendix A).

Using these three gene lists, key pathways were identified, and a PPI network was constructed by intersecting all genes from the key pathways with ISO targets. Candidate therapeutic targets were identified using the MCODE [36] algorithm. Glycolysis- and phagocytosis-related genes were retrieved from the GeneCards [33] and MSigDB databases [37], and candidate therapeutic targets were further intersected with these phenotype-related genes. Subsequently, eight machine learning methods [38,39,40,41,42] were applied to identify phenotype-specific key therapeutic targets.

Clinical diagnostic evaluations—including nomogram analysis, receiver operating characteristic (ROC) analysis, correlation analysis, and intergroup expression comparisons—were conducted on the key phenotype-related targets. GSEA was also performed to explore the functional roles of these targets. Immune cell analysis of the mRNA expression cohort from COPD AM was conducted to assess the associations between phenotype-related key targets and immune cell infiltration. Molecular docking simulations were performed to predict interactions between ISO and the identified phenotype-related key targets. Finally, the findings were validated in an independent dataset. The overall workflow of the study is summarized in Figure 9.

### 4.2. Data Description

The training set of this study involved 46 AM samples obtained from bronchoalveolar lavage fluid, including 22 patients with COPD and 24 healthy non-smoking controls, and the validation set also involved AM samples obtained from bronchoalveolar lavage fluid, including 12 COPD patients and 24 healthy non-smoking controls (Table 3, Appendix A). Microarray data were obtained from the GPL570 platform (Affymetrix Human Genome U133 Plus 2.0 Array). The training dataset and validation dataset were previously published in the GEO database (https://www.ncbi.nlm.nih.gov/geo/ (accessed on 2 November 2024)) under accession numbers GSE130928 [43] and GSE13896 [44], respectively. Genes related to phagocytosis and specific phenotypes were retrieved from the GeneCards database [33] (https://www.genecards.org/ (accessed on 2 November 2024)) and the MSigDB database [37] (https://www.gsea-msigdb.org/gsea/index.jsp (accessed on 2 November 2024)).

### 4.3. Identification of DEGs in COPD Patients

Differential expression analysis was conducted using the limma package in R software (version 4.2.1). Genes with a threshold of |logFC| > 0.25 and an adjusted *p*-value < 0.05 were identified, and the results were visualized using a volcano plot [45].

### 4.4. Screening of Potential Target Genes Using WGCNA

The top 5414 differentially genes were extracted from the AM microarray data of COPD patients and healthy controls and analyzed using the WGCNA package in R software (version 4.2.1) [46]. A co-expression network was constructed and transformed into a topological overlap matrix (TOM) through hierarchical clustering to identify gene modules and calculate module eigengenes. The correlation between each gene module and COPD or normal samples was assessed to identify key modules associated with COPD. Genes from the key modules were then intersected with the DEGs using the Venn package in R software (version 4.2.1) to identify potential target genes for COPD.

### 4.5. Identification of Predicted Targets of ISO and COPD

The predicted molecular targets of ISO were retrieved from the TCMSP database [34] (https://old.tcmsp-e.com/tcmsp.php (accessed on 2 November 2024)), the CTD [32] (https://ctdbase.org/ (accessed on 2 November 2024)), and PubChem [35] (https://pubchem.ncbi.nlm.nih.gov/ (accessed on 2 November 2024)). Similarly, COPD-related targets were identified using the CTD [32] (http://ctdbase.org/ (accessed on 2 November 2024)) and GeneCards [33] (https://www.genecards.org/ (accessed on 2 November 2024)) databases. The overlapping targets of ISO and COPD were determined using the Venn package in R software (version 4.2.1), which identified potential shared targets for further analysis.

### 4.6. Pathway Enrichment Analysis and Identification of Predicted Therapeutic Target Genes

To identify potential target genes for COPD, the Venn package in R software (version 4.2.1) was used to intersect the results of differential expression analysis from microarray data with the results of WGCNA analysis. ISO-related target genes were obtained from three databases: TCMSP, CTD, and PubChem. COPD-related target genes were identified from disease-specific databases, and the intersection of these datasets was used to determine predicted target genes for COPD. Pathway enrichment analysis was performed on these target genes using the clusterProfiler package in R software (version 4.2.1), identifying key pathways categorized as KEGGa, KEGGb, and KEGGc according to the KEGG database (https://www.genome.jp/kegg/ (accessed on 11 February 2025)) [47]. The Venn package was employed to identify overlapping pathways among these three categories. Finally, all genes associated with the key pathways, as annotated in the KEGG database, were intersected with the ISO-predicted target genes to identify potential therapeutic target genes for COPD.

### 4.7. Construction of PPI Networks and Small Molecule–Pathway–Target Networks

To investigate the interactions among the predicted therapeutic target genes, a PPI network was constructed using the STRING database [48] (https://string-db.org/ (accessed on 11 February 2025)), with a minimum interaction score threshold of 0.4 to identify significant interactions. The resulting PPI network was visualized using Cytoscape software, version 3.10.2 [49]. To identify key subnetworks within the PPI network, the MCODE plugin in Cytoscape was employed for module analysis, highlighting critical clusters of protein interactions [36].

The genes identified in the PPI network were further intersected with glycolysis- and phagocytosis-related phenotype genes using the Venn package in R software (version 4.2.1), enabling the identification of phenotype-related candidate therapeutic targets.

Additionally, a small molecule–pathway–target network was constructed and visualized in Cytoscape. This network integrates ISO, its predicted target genes, and the key pathways enriched with these target genes, providing a comprehensive representation of the molecular mechanisms and therapeutic potential of ISO in the treatment of COPD.

### 4.8. Key Goals for Screening COPD Patients Using Machine Learning Algorithms

In this study, eight machine learning algorithms were applied to identify phenotype-related key therapeutic targets for ISO treatment in COPD patients. The specific algorithms included the following: Least Absolute Shrinkage and Selection Operator (Lasso) [38], Support Vector Machine (SVM) [38], Random Forest (RF) [38], eXtreme Gradient Boosting (XGBoost) [38], Bayesian [41], Treebag [39], Boruta [40], and Learning Vector Quantization (LVQ) [42]. These algorithms were implemented using the following R software (version 4.4.1) packages: glmnet, e1071, xgboost, caret, and Boruta. The genes identified from this intersection were considered the final key therapeutic targets for ISO’s treatment of COPD.

To further validate these targets, the Rms package was utilized to construct bar graphs reflecting potential therapeutic markers. ROC analysis was performed using AM mRNA microarray data to evaluate the clinical diagnostic value of these key genes. The pROC package was employed to calculate the Area Under the Curve (AUC) values for these genes, providing a quantitative assessment of their diagnostic performance [50].

### 4.9. Characterization of Key Targets for Expression, Correlation, and Gene Set Enrichment Analysis

To investigate the expression and functional characteristics of key therapeutic targets, the corrplot package in R software (version 4.2.1) was utilized to assess and visualize the correlation between these targets. The Wilcoxon rank-sum test was employed to analyze the expression levels of each key gene in different groups, providing statistical validation for their differential expression. Subsequently, GSEA was performed for each key gene to identify the biological pathways associated with the targets.

### 4.10. Immune Cell Analysis

Considering the critical role of the immune system in the pathophysiology of COPD, immune cell analysis was conducted using mRNA microarray data from AM. The analysis was performed using the CIBERSORT package, which deconvolutes mRNA expression data into the proportions of 22 different immune cell types [51]. This computational approach effectively determined the immune cell composition in numerous samples and allowed for the association of potential therapeutic markers with specific immune cell populations.

### 4.11. Molecular Docking Validation

To validate the interaction between ISO and its predicted key targets, molecular docking was conducted. The crystal structure of the main targets was retrieved from the UniProt database [52] (https://www.uniprot.org/ (accessed on 20 February 2025)), with PDB files 1H10, 1EKU, and 1A02 used as structural inputs.

The 3D structure of ISO was obtained from the PubChem database [35] (https://pubchem.ncbi.nlm.nih.gov/ (accessed on 20 February 2025)). Molecular docking was performed using the CB-Dock2 online platform [53] (https://cadd.labshare.cn/cb-dock2/index.php (accessed on 20 February 2025)), a tool designed to identify the binding cavities of a target protein and predict binding affinities. Each docking experiment was repeated five times, and the lowest binding affinity (indicative of the strongest binding interaction) was recorded for each target-small molecule pair.

### 4.12. Statistical Analysis

The data were analyzed statistically using R software (version 4.2.1). A significance threshold of *p* < 0.05 was established, and all tests were two-tailed. Comparisons between the two groups were conducted using a *t*-test (assuming normality and homogeneity of variance), Welch’s *t*-test (assuming normality but not homogeneity of variance), or the Wilcoxon rank-sum test (non-normally distributed data, nonparametric test). Correlation analyses were performed using Pearson or Spearman correlation coefficients, depending on the data characteristics.

## 5. Conclusions

Using bioinformatics approaches, we identified and validated key targets of ISO in regulating glycolysis and phagocytosis in COPD, specifically *AKT1*, *IFNG*, and *JUN*. These findings provide a theoretical foundation for clinical treatment strategies and future drug development.

## Figures and Tables

**Figure 1 ijms-26-03907-f001:**
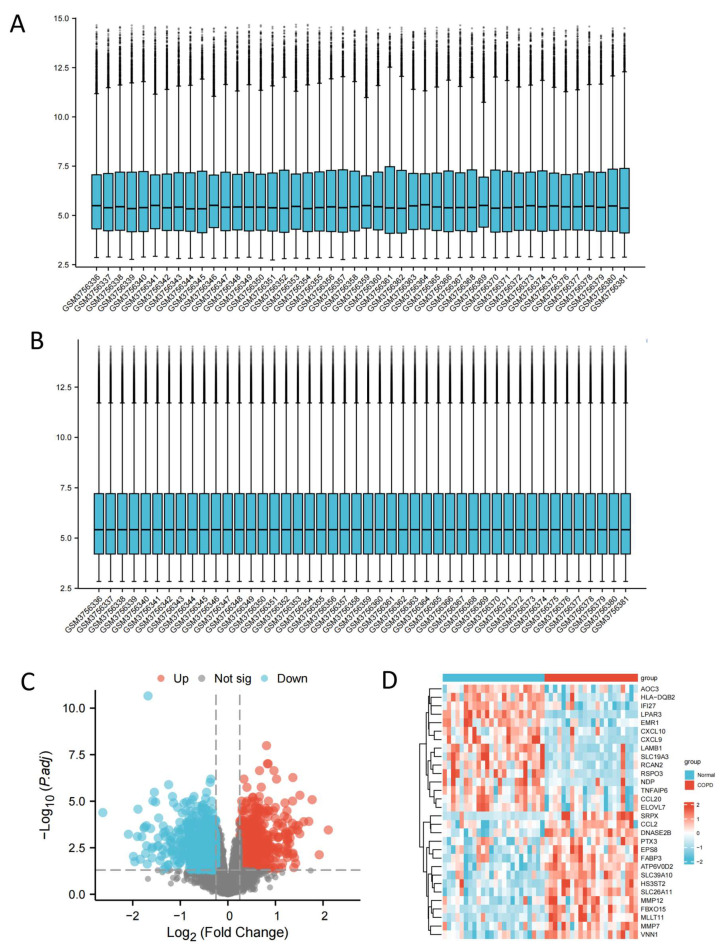
Identification of DEGs in the AM of COPD patients. (**A**) Box plot before standardization. (**B**) Box plot after standardization. (**C**) Volcano plot showing differentially expressed genes in COPD; red indicates significantly upregulated genes, and blue indicates significantly downregulated genes. (**D**) Heat map showing the expression of the top 30 differentially expressed genes in the sample.

**Figure 2 ijms-26-03907-f002:**
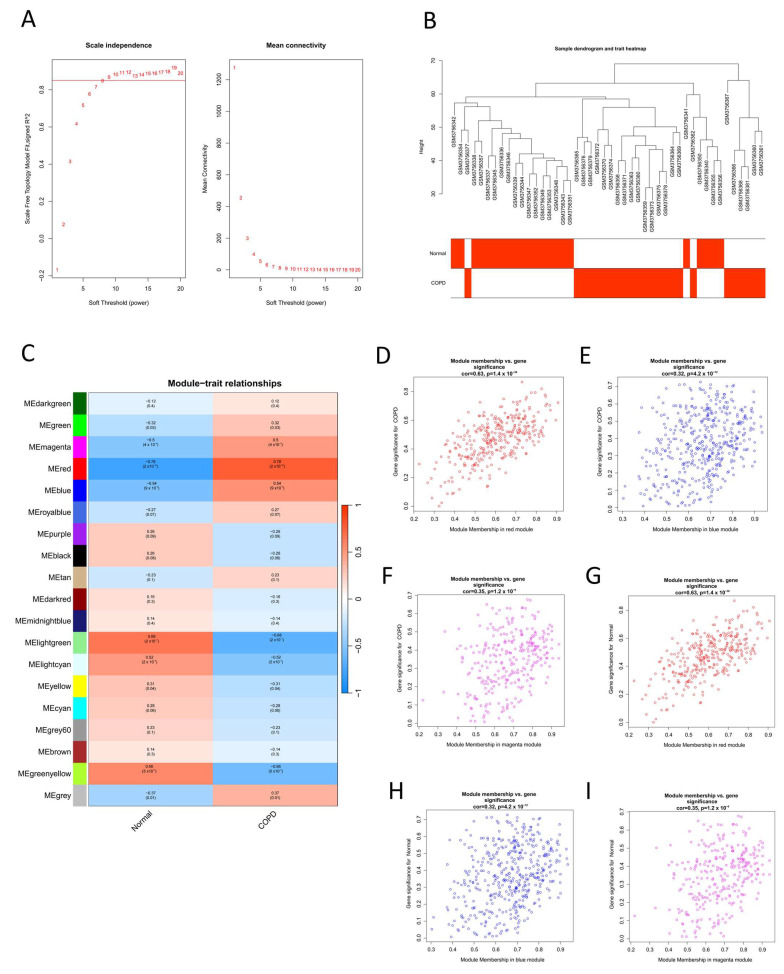
Construction of a weighted gene co-expression network for COPD. (**A**) Soft threshold β = 9 and scale-free topological fit index (R^2^). (**B**) Dendrogram of sample clustering, with each sample corresponding to a leaf. (**C**) Heat map of module–trait correlation. Of note, 18 rows correspond to each of the 18 combined modules, 2 columns correspond to the normal group and the COPD group, and the cells contain the corresponding correlation coefficients and *p*-values. (**D**–**F**) Scatter plots of MM-GS for red, blue, and magenta modules in the COPD group. (**G**–**I**) Scatter plots of MM-GS for red, blue, and magenta modules in the control group. The above graphics were drawn using R software (version 4.2.1).

**Figure 3 ijms-26-03907-f003:**
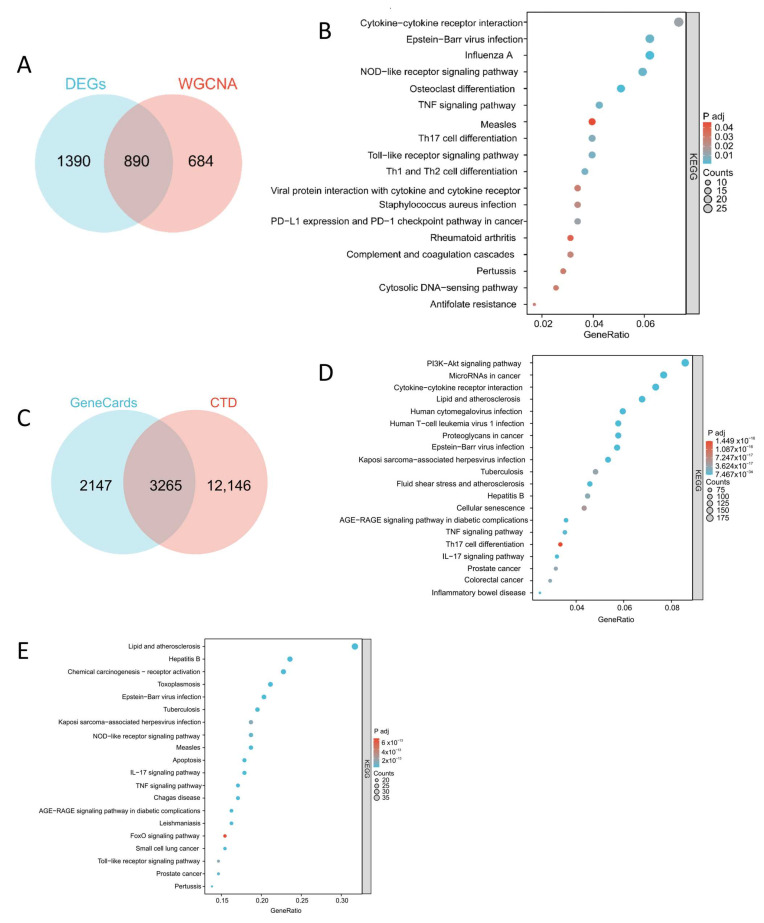
Identification of key KEGG pathways. (**A**) DEGs mapped to WGCNA key module genes. (**B**) KEGGa, KEGG enrichment results of genes intersecting DEGs with WGCNA key module genes. (**C**) Cross-results of genes predicted using the CTD and GeneCards databases. (**D**) KEGGb, KEGG enrichment results of COPD-related genes in the disease databases. (**E**) KEGGc, KEGG enrichment results of ISO target genes. The above graphics were drawn using R software (version 4.2.1).

**Figure 4 ijms-26-03907-f004:**
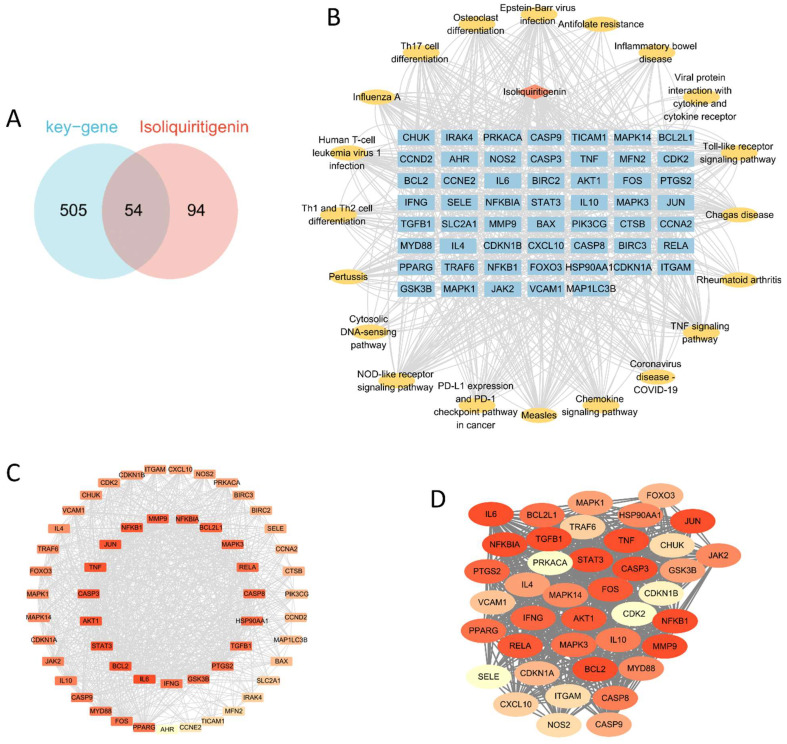
Network construction of predicted therapeutic target genes. (**A**) The intersection of ISO target genes and all genes in the key KEGG pathway. (**B**) The “component-target-pathway” network, in which the blue rectangular nodes are the 54 predicted therapeutic target genes, the light red diamond nodes are ISO, and the yellow oval nodes are key KEGG pathways. (**C**) The PPI network constructed using the 54 predicted therapeutic target genes. (**D**) The key network obtained based on MCODE. In (**C**,**D**), a redder node color indicates a greater number of interactions with other genes.

**Figure 5 ijms-26-03907-f005:**
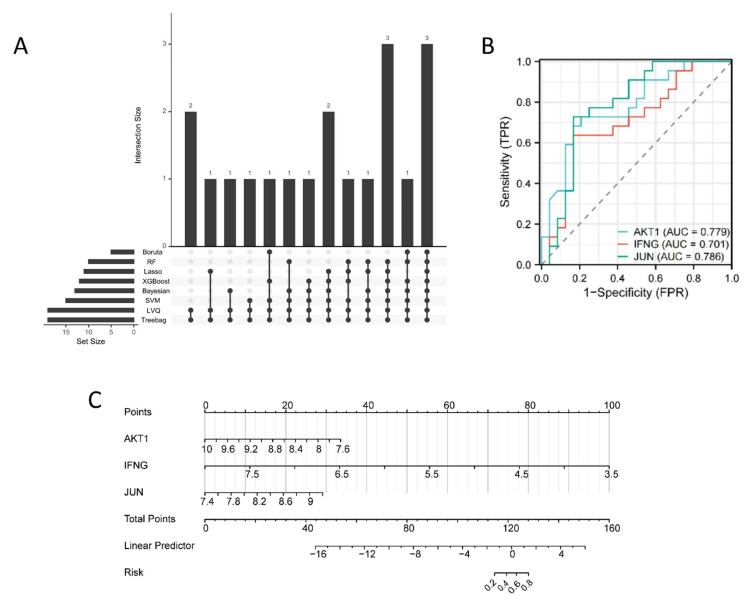
Key target screening based on machine learning. (**A**) Screening of key targets based on eight machine learning algorithms, including Boruta, Random Forest (RF), Least Absolute Shrinkage and Selection Operator (Lasso), eXtreme Gradient Boosting (XGBoost), Bayesian, Support Vector Machine (SVM), Learning Vector Quantization (LVQ), and Tree-bag. (**B**) ROC curve to evaluate the diagnostic efficacy of the key targets. (**C**) Nomogram of key targets to evaluate COPD risk.

**Figure 6 ijms-26-03907-f006:**
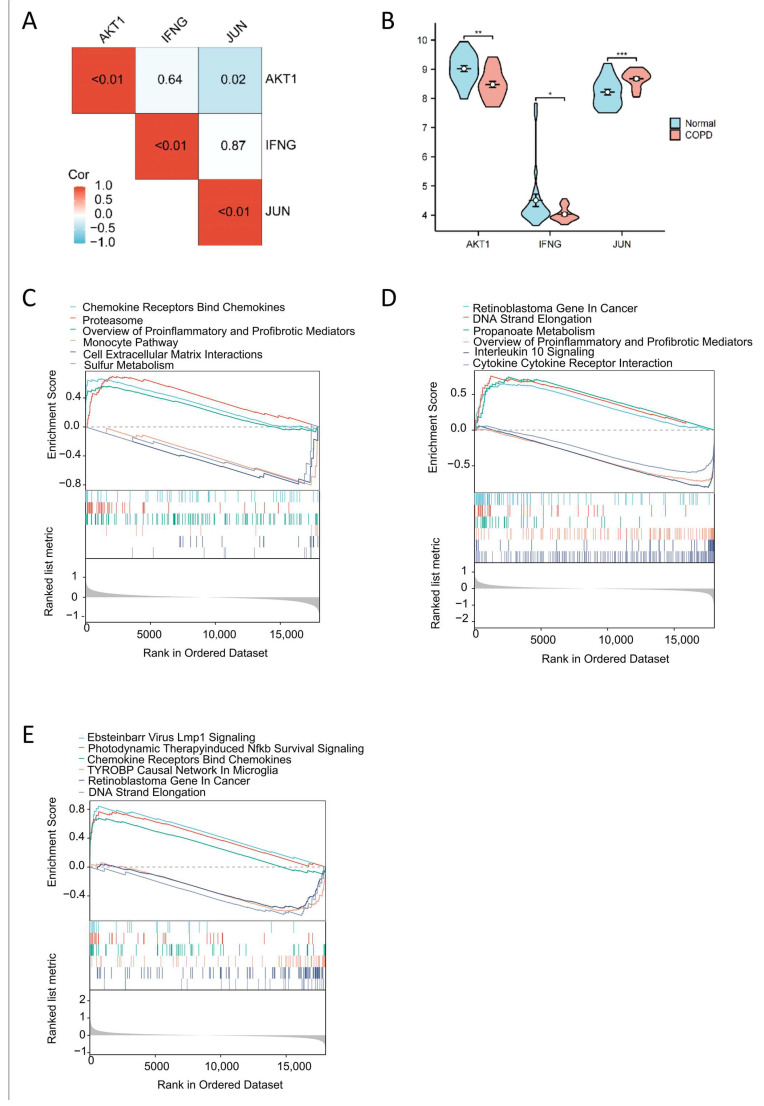
Correlation analysis, expression characterization, and GSEA enrichment results of key targets. (**A**) Correlation heat map of key targets; the numbers in the box are *p*-values. (**B**) Expression of key genes in AM mRNA microarray data, with blue for the healthy group and pink for the COPD group. (**C**–**E**) GSEA enrichment results of *AKT1*, *IFNG*, and *JUN*, respectively. *: *p* < 0.05; **: *p* < 0.01; ***: *p* < 0.001.

**Figure 7 ijms-26-03907-f007:**
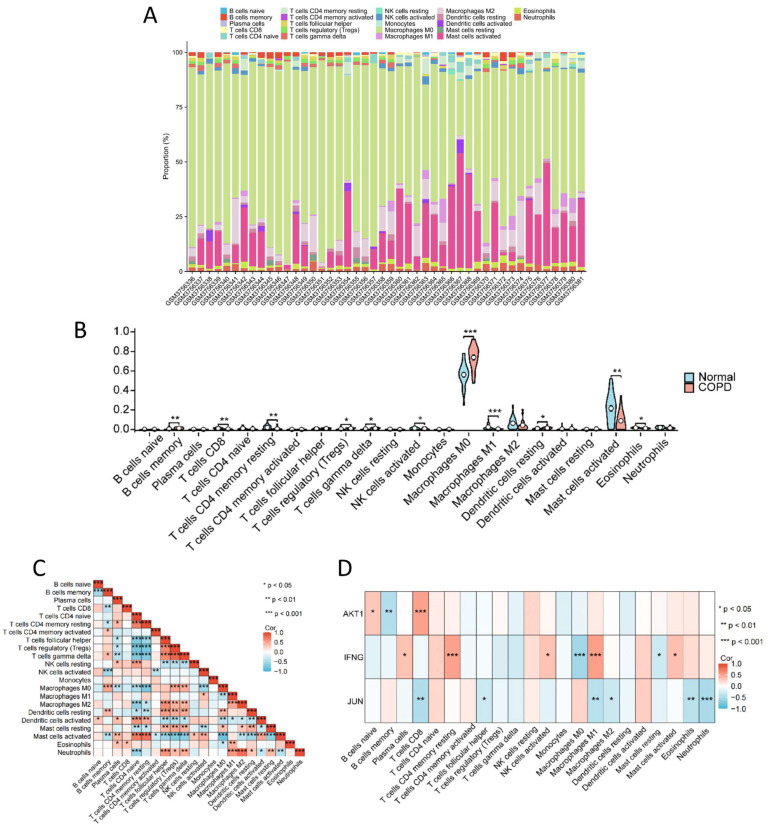
Immune cell analysis. (**A**) Relative proportions of the 22 immune cell subsets in all samples. (**B**) Differences in the levels of the 22 immune cells between the normal group and the COPD group (*: *p* < 0.05, **: *p* < 0.01, and ***: *p* < 0.001). (**C**) Correlation between the 22 immune cell subsets. (**D**) Correlation between key targets and the 22 immune cell subsets.

**Figure 8 ijms-26-03907-f008:**
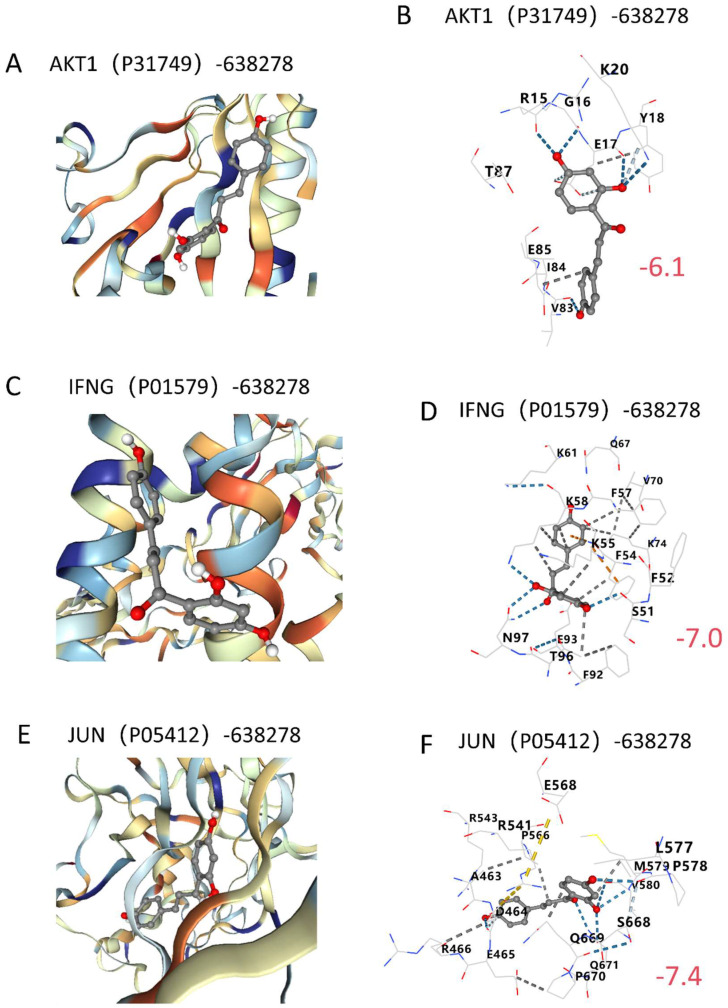
Molecular docking validation. (**A**,**C**,**E**) Visualization of the docking model with binding affinity. (**B**,**D**,**F**) Detailed diagram of the docking model, with red numbers indicating binding affinity.

**Figure 9 ijms-26-03907-f009:**
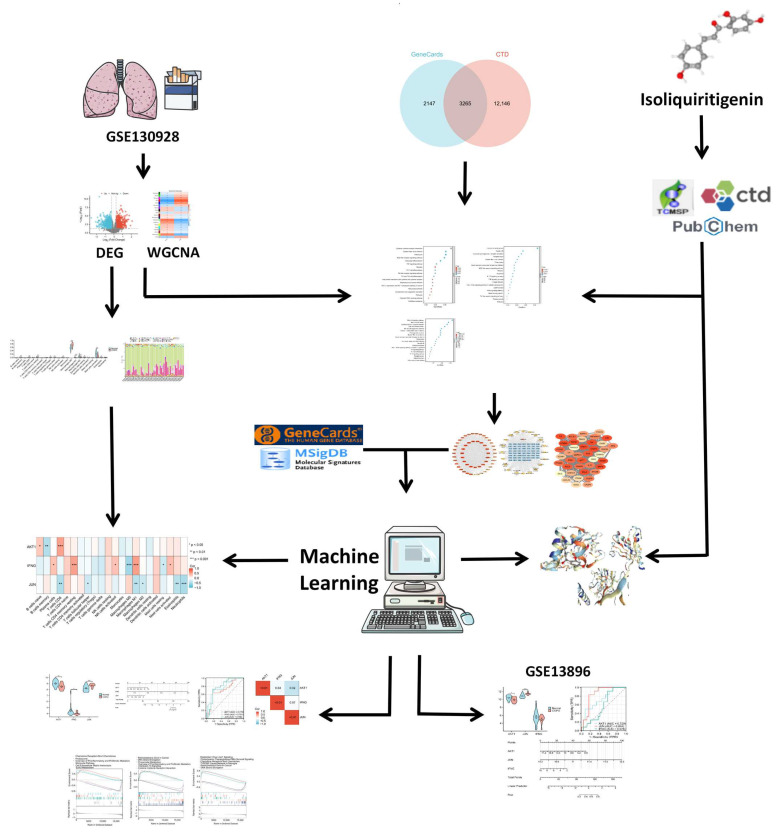
Graphical workflow of this study. We conducted differential expression analysis and WGCNA on the mRNA expression dataset of COPD patients (GSE130928). COPD-related genes were sourced from the CTD and GeneCards. ISO-related targets were obtained from the TCMSP, CTD, and PubChem databases. Key pathways were identified from these genes, and those within the key pathways were extracted and intersected to construct a PPI network. Candidate therapeutic targets were identified using the MCODE algorithm and further intersected with genes related to glycolysis and phagocytosis. To identify phenotype-specific therapeutic targets, we applied eight machine learning methods. Clinical evaluations of the key targets included nomogram analysis, ROC analysis, GSEA, immune cell analysis, and molecular docking simulations. The findings were validated in an independent dataset (GSE13896). *: *p* < 0.05; **: *p* < 0.01; ***: *p* < 0.001.

**Table 1 ijms-26-03907-t001:** Number of enriched genes and *p*-values of 20 key KEGG pathways.

KEGG Pathways	KEGGa Count	KEGGa *p*-Value	KEGGb Count	KEGGb *p*-Value	KEGGc Count	KEGGc *p*-Value	SUM. Count	AVG. *p*-Value
Influenza A	22	0.001	94	<0.001	19	<0.001	135	<0.001
Osteoclast differentiation	18	0.001	71	<0.001	15	<0.001	104	<0.001
NOD-like receptor signaling pathway	21	0.004	81	<0.001	23	<0.001	125	0.001
Epstein–Barr virus infection	22	0.004	120	<0.001	25	<0.001	167	0.001
TNF signaling pathway	15	0.005	74	<0.001	21	<0.001	110	0.002
Toll-like receptor signaling pathway	14	0.007	63	<0.001	18	<0.001	95	0.0023
Th1 and Th2 cell differentiation	13	0.007	52	<0.001	12	<0.001	77	0.0023
Th17 cell differentiation	14	0.008	70	<0.001	17	<0.001	101	0.003
PD-L1 expression and PD-1 checkpoint pathway in cancer	12	0.013	53	<0.001	16	<0.001	81	0.0043
Viral protein interaction with cytokine and cytokine receptor	12	0.029	65	<0.001	4	<0.001	81	0.0397
Cytosolic DNA-sensing pathway	9	0.029	24	<0.001	6	<0.001	39	0.021
Pertussis	10	0.029	51	<0.001	17	<0.001	78	0.01
Antifolate resistance	6	0.029	17	<0.001	5	<0.001	28	0.011
Rheumatoid arthritis	11	0.039	57	<0.001	6	<0.001	74	0.015
Measles	14	0.046	77	<0.001	23	<0.001	114	0.015
Chemokine signaling pathway	17	0.064	102	<0.001	14	<0.001	133	0.022
Chagas disease	11	0.068	63	<0.001	21	<0.001	95	0.023
Coronavirus disease—COVID-19	19	0.081	102	<0.001	16	<0.001	137	0.027
Inflammatory bowel disease	8	0.092	52	<0.001	10	<0.001	70	0.031
Human T-cell leukemia virus 1 infection	18	0.099	121	<0.001	21	<0.001	160	0.033

**Table 2 ijms-26-03907-t002:** Key targets, selected protein IDs, and docking pockets.

Gene	Uniprot ID	PDB ID	Macromolecule	Center X	Center Y	Center Z	Size X	Size Y	Size Z
*AKT1*	P31749	1H10	AKT Serine/Threonine Kinase 1	23	19	22	22	22	22
*IFNG*	P01579	1EKU	Interferon Gamma	52	49	52	22	22	22
*JUN*	P05412	1A02	Jun Proto-Oncogene, AP-1 Transcription Factor Subunit	25	21	60	35	22	22

**Table 3 ijms-26-03907-t003:** GEO dataset information.

GEO Datasets	Normal Non-Smoking/Case	COPD/Case	Species	Method	Organization Source	Type	Platform
GSE130928 [43]	24	22	*Homo sapiens*	Bronchoalveolar lavage	Alveolar macrophages	Expression profiling by array	GPL570
GSE13896 [44]	24	12	*Homo sapiens*	Bronchoalveolar lavage	Alveolar macrophages	Expression profiling by array	GPL570

## Data Availability

The datasets analyzed during the current study are available in the GEO database (https://www.ncbi.nlm.nih.gov/geo/ (accessed on 2 November 2024)), GeneCards database (https://www.genecards.org/ (accessed on 2 November 2024)), MSigDB database (https://www.gsea-msigdb.org/gsea/msigdb/index.jsp (accessed on 2 November 2024)), TCMSP database (https://old.tcmsp-e.com/tcmsp.php (accessed on 2 November 2024)), the CTD (https://ctdbase.org/ (accessed on 2 November 2024)), PubChem (https://pubchem.ncbi.nlm.nih.gov/ (accessed on 20 February 2025)), and UniProt database (https://www.uniprot.org/ (accessed on 20 February 2025)).

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
