# Peer review of "Bioinformatics Approach to Identifying Molecular Targets of Isoliquiritigenin Affecting Chronic Obstructive Pulmonary Disease: A Machine Learning Pharmacology Study"

_ijms, 2025, doi:10.3390/ijms26083907_

Round 1

Reviewer 1 Report

Comments and Suggestions for Authors

Manuscript entitled:“Identification of molecular targets of isoliquiritigenin affecting COPD by regulating glycolysis and phagocytosis in alveolar macrophages: a machine learning pharmacology study” by Sha Huang, Lulu Zhang, and Xiaoju Liu.

This manuscript presents an interesting machine learning-based pharmacological study investigating the molecular targets of isoliquiritigenin (ISO) in regulating glycolysis and phagocytosis in alveolar macrophages in the context of chronic obstructive pulmonary disease (COPD). The study integrates bioinformatics approaches, gene expression data analysis, and molecular docking simulations to identify key targets, including AKT1, IFNG, and JUN. The research is timely, as COPD remains a major global health burden, and new therapeutic strategies are needed.

The manuscript is well-organized and methodologically sound, but several aspects require clarification and improvements in both scientific rigor and presentation.

Comments:

  1. The introduction provides a solid background on COPD, glycolysis, and phagocytosis but does not sufficiently elaborate on the role of isoliquiritigenin (ISO).
  2. It would be beneficial to provide a more detailed discussion of ISO’s pharmacological effects, especially in the context of immune modulation and glycolysis regulation.
  3. The choice of datasets (GSE130928 and GSE13896) should be better justified. Are there any limitations to these datasets, such as small sample size or patient heterogeneity?
  4. Provide a table summarizing patient characteristics in the datasets, including disease severity, comorbidities, and smoking history.
  5. The docking results suggest strong binding of ISO to the targets, but experimental validation (e.g., in vitro assays) is lacking.
  6. Were any of the predicted changes in immune cell populations confirmed in patient blood samples or bronchoalveolar lavage fluid (BALF)?
  7. Was clinical COPD severity (e.g., GOLD classification) correlated with the expression levels of AKT1, IFNG, and JUN?
  8. The discussion appropriately links findings to prior research, but it should address potential confounders such as medication use and dietary intake, which could influence gene expression.
  9. The authors should compare their findings to previous studies on ISO in lung diseases. Are there discrepancies?
  10. The therapeutic potential of ISO should be further elaborated. Are there existing COPD models where ISO has been tested?
  11. Limitations should be expanded, especially regarding the reliance on in silico methods without experimental validation.

Author Response

1- The introduction provides a solid background on COPD, glycolysis, and phagocytosis but does not sufficiently elaborate on the role of isoliquiritigenin (ISO).

Reply: Thank you very much for this question, and we have made the following changes in the manuscript:

Isoliquiritigenin (ISO) is a flavonoid compound and one of the most biologically active chalcones derived from the roots of licorice and other plants9. ISO demonstrates therapeutic potential against gastric cancer by decreasing GLUT4-mediated glucose uptake and inhibiting the protein expression of c-Myc and HIF-1α, ultimately inducing a collapse of energy metabolism10. In colorectal cancer, ISO exerts its effects by inhibiting glycolysis and lactate production through the modulation of AMPK and the Akt/mTOR signaling pathways11. Furthermore, in non-tumor diseases, ISO enhances the phagocytic activity of immune cells by regulating SLC7A11-mediated glycolysis, thereby facilitating wound healing in diabetic conditions12. ISO also enhance the phagocytosis of RAW264.7 treated with a toxic substance (BDE-47) and alleviate the immune function damage caused by BDE-47 13. Furthermore, ISO demonstrated protective effects against cigarette smoke-induced COPD in mouse model 14.

2- It would be beneficial to provide a more detailed discussion of ISO’s pharmacological effects, especially in the context of immune modulation and glycolysis regulation.

Reply: Thank you very much for this question, and we have made the following changes in the manuscript:

Isoliquiritigenin (ISO) is a flavonoid compound and one of the most biologically active chalcones derived from the roots of licorice and other plants9. ISO demonstrates therapeutic potential against gastric cancer by decreasing GLUT4-mediated glucose uptake and inhibiting the protein expression of c-Myc and HIF-1α, ultimately inducing a collapse of energy metabolism10. In colorectal cancer, ISO exerts its effects by inhibiting glycolysis and lactate production through the modulation of AMPK and the Akt/mTOR signaling pathways11. Furthermore, in non-tumor diseases, ISO enhances the phagocytic activity of immune cells by regulating SLC7A11-mediated glycolysis, thereby facilitating wound healing in diabetic conditions12. ISO also enhance the phagocytosis of RAW264.7 treated with a toxic substance (BDE-47) and alleviate the immune function damage caused by BDE-47 13. Furthermore, ISO demonstrated protective effects against cigarette smoke-induced COPD in mouse model 14.

3- The choice of datasets (GSE130928 and GSE13896) should be better justified. Are there any limitations to these datasets, such as small sample size or patient heterogeneity?

Reply: Thank you very much for this suggestion, and we have made the following changes in the manuscript:

Second, due to the specific focus on AM in COPD, there were limited options for public datasets. While we utilized GSE130928 and GSE13896 as training and validation sets to enhance the accuracy of our findings, the sample sizes analyzed were relatively small and may exhibit heterogeneity. Therefore, additional sequencing data with a larger sample size or further experimental validation is necessary to support the findings of our study.

4- Provide a table summarizing patient characteristics in the datasets, including disease severity, comorbidities, and smoking history.

Reply: Thank you for your suggestion. We have added the disease severity and smoking history of the patients and other patient characteristics in the supplementary materials. Since the original article did not include comorbidities, this content was not added. The supplementary content and data are from the published literature and have been noted in the table.

5- The docking results suggest strong binding of ISO to the targets, but experimental validation (e.g., in vitro assays) is lacking.

Reply: Thank you for highlighting this issue; we have included this limitation in the limitations section of the study.

This study has several limitations. First, it is based on publicly available datasets and database information, utilizing computational methods that have not undergone experimental validation (e.g., human, animal, in vitro, etc.). However, we have corroborated our findings using external datasets. Future research will aim to confirm these results through in vitro and/or in vivo experiments.

6- Were any of the predicted changes in immune cell populations confirmed in patient blood samples or bronchoalveolar lavage fluid (BALF)?

Reply: Thank you for highlighting this issue; we have included this limitation in the limitations section of the study.

This study has several limitations. First, it is based on publicly available datasets and database information, utilizing computational methods that have not undergone experimental validation (e.g., human, animal, in vitro, etc.). However, we have corroborated our findings using external datasets. Future research will aim to confirm these results through in vitro and/or in vivo experiments.

7- Was clinical COPD severity (e.g., GOLD classification) correlated with the expression levels of AKT1, IFNG, and JUN?

Reply: Thank you for raising this question, but since the authors who submitted the data did not submit the disease severity data to the public database, we cannot judge whether the severity of clinical COPD is associated with the expression levels of AKT1, IFNG and JUN based on the public database. We have not conducted experimental verification in the human population, so this is also our limitation. We have added this limitation in the limitation section.

This study has several limitations. First, it is based on publicly available datasets and da-tabase information, utilizing computational methods that have not undergone experimental validation (e.g., human, animal, in vitro, etc.). However, we have corroborated our findings using external datasets. Future research will aim to confirm these results through in vitro and/or in vivo experiments…

…Finally, the public databases included in this study lack critical clinical information, including lung function, disease severity classification, comorbidities, medication use, and dietary intake. This absence hinders the ability to correlate the identified molecules with clinical data, and it also prevents the adjustment for potential confounding factors that may influence the results.

8- The discussion appropriately links findings to prior research, but it should address potential confounders such as medication use and dietary intake, which could influence gene expression.

Reply: Thank you for raising this question. Since the authors who submitted the data did not collect or submit data such as medication use and dietary intake to public databases, we cannot adjust these potential confounders based on public databases, so this is also our limitation. We have added this limitation in the limitation section. However, we have verified it in external datasets.

Finally, the public databases lack critical clinical information, including lung function, disease severity classification, comorbidities, medication use, and dietary intake. This absence hinders the ability to correlate the identified molecules with clinical data, and it also prevents the adjustment for potential confounding factors that may influence the results.

9- The authors should compare their findings to previous studies on ISO in lung diseases. Are there discrepancies?

Reply: Thank you very much for this suggestion, and we have made the following changes in the manuscript:

This study suggests that ISO may be beneficial in the treatment of COPD. These findings align with those of Yu et al., who demonstrated that ISO mitigated lung pathological damage in a COPD mouse14. Furthermore, they reported that ISO exerts protective effects in COPD mice by modulating inflammation and oxidative stress, specifically through the modulation of myeloperoxidase activity, malondialdehyde levels, and the expression and activity of key signaling pathways, including nuclear factor erythroid 2-related factor 2 (Nrf2) and nuclear factor κB (NF-κB) 14. ISO has also demonstrated therapeutic effects in other respiratory conditions, including acute lung injury and lung cancer 26,27.

10- The therapeutic potential of ISO should be further elaborated. Are there existing COPD models where ISO has been tested?

Reply: Thank you very much for this suggestion, and we have made the following changes in the manuscript:

This study suggests that ISO may be beneficial in the treatment of COPD. These findings align with those of Yu et al., who demonstrated that ISO mitigated lung pathological damage in a COPD mouse14. Furthermore, they reported that ISO exerts protective effects in COPD mice by modulating inflammation and oxidative stress, specifically through the modulation of myeloperoxidase activity, malondialdehyde levels, and the expression and activity of key signaling pathways, including nuclear factor erythroid 2-related factor 2 (Nrf2) and nuclear factor κB (NF-κB) 14. ISO has also demonstrated therapeutic effects in other respiratory conditions, including acute lung injury and lung cancer 26,27.

11- Limitations should be expanded, especially regarding the reliance on in silico methods without experimental validation.

Reply: Thank you very much for this suggestion, and we have made the following changes in the manuscript:

This study has several limitations. First, it is based on publicly available datasets and database information, utilizing computational methods that have not undergone experimental validation (e.g., human, animal, in vitro, etc.). However, we have corroborated our findings using external datasets. Future research will aim to confirm these results through in vitro and/or in vivo experiments. Second, due to the specific focus on AM in COPD, there were limited options for public datasets. While we utilized GSE130928 and GSE13896 as training and validation sets to enhance the accuracy of our findings, the sample sizes analyzed were relatively small and may exhibit heterogeneity. Therefore, additional sequencing data with a larger sample size or further experimental validation is necessary to support the findings of our study. Finally, the public databases included in this study lack critical clinical information, including lung function, disease severity classification, comorbidities, medication use, and dietary intake. This absence hinders the ability to correlate the identified molecules with clinical data, and it also prevents the adjustment for potential confounding factors that may influence the results.

Reviewer 2 Report

Comments and Suggestions for Authors

In this study, Huang et.al have taken 2 publicly available RNA expression datasets (GSE130928 and GSE13896 ) for COPD from the GEO database and reanalyzed these datasets with guidance from GeneCards and MSigDB databases specific to genes related to glycolysis and phagocytosis. The COPD-specific glycolysis and phagocytosis gene signatures were intersected with the ISO target genes obtained from TCMSP, CTD and PubChem databases. This newly generated gene list was analyzed for gene ontology GSEA and PPI mapping.  This paper used exclusively a bioinformatic approach without any wet-lab experimental evaluation to validate and confirm the findings. Therefore, whatever is presented in this paper is only hypothetical. That is a big weakness. Keeping this mind, the authors should change the title as “Bioinformatic approach to identifying molecular targets of isoliquiritigenin affecting COPD: a machine learning pharmacology study”.  This is because there’s no experimental evidence provided in this paper to show that the targets of ISO regulates glycolysis and phagocytosis of AM.

Other issues are:

Line 65-66 is incomplete.

Line 78. Describe the source data details here. It is unclear what was the nature of the original study from which the AM-RNAchip data was taken to use here.

Change the order of Figure numbers. Right now, it starts with Figure-2.  

Line 120. Explain what is gene ratio in figure 4?

Figure-3: Mention what software was used to obtain the figures? And mention what is the sample size (n) of normal and COPD in figure 3C?

Figure-4: Mention what software was used to obtain the figures?

Line 129. What was the source of isoliquiritigenin target genes ? Mention it here.

Line 134. Start the word with a capital letter A.

Figure-5. Explain what is the meaning for each of different colors (red, blue, etc) in the legend?.

Table-1. Mention the source of isoliquiritigenin dataset here. Also, include a column to show the references of the GEO dataset taken for analysis in this paper.

Line 353. Revise the figure number and briefly describe summary of workflow chart in the legend of Figure-1.

In the Methods Section, include a paragraph on statistical analysis methods applied of various datasets  and how significance was established for each of them.

Line 460-470. Move this Conclusion section to the last paragraph of Discussion section.

Comments on the Quality of English Language

Need to check thoroughly for spelling and grammatical errors.

Author Response

1- Keeping this mind, the authors should change the title as “Bioinformatic approach to identifying molecular targets of isoliquiritigenin affecting COPD: a machine learning pharmacology study”. 

Reply: Thank you very much for this suggestion and we have made changes in the manuscript as follows:

Bioinformatic approach to identifying molecular targets of isoliquiritigenin affecting COPD: a machine learning pharmacology study

2- Line 65-66 is incomplete.

Reply: Thank you very much for this suggestion, and we have made the following changes in the manuscript:

We speculate that ISO may also affect COPD by affecting the glycolysis and phagocytic function of COPD, but the possible target of action is still unclear. Therefore, this study aims to investigate the molecular targets through which ISO influences COPD by regulating glycolysis and phagocytosis.

3- Line 78. Describe the source data details here. It is unclear what was the nature of the original study from which the AM-RNAchip data was taken to use here.

Reply: Thank you very much for this suggestion, and we have made changes in the manuscript.

To identify genes involved in COPD pathophysiology, AM RNA chip data were normalized and subjected to differential expression analysis. By applying the screening criteria of |logFC| > 0.25 and adjP < 0.05, we identified a total of 2,280 differentially expressed genes (DEGs) in the GSE130928 microarray dataset, comprising 1,108 upregulated genes and 1,172 downregulated genes.

The training set of this study involved 46 AM samples obtained from bronchoalveolar lavage fluid, including 22 patients with COPD and 24 healthy non-smoking controls, and the validation set also involved AM samples obtained from bronchoalveolar lavage fluid, including 12 COPD patients and 24 healthy non-smoking controls (Table 3, Table S4, Table S5). Microarray data were obtained from the GPL570 platform (Affymetrix Human Genome U133 Plus 2.0 Array). The training dataset and validation dataset were previously published in the GEO database (https://www.ncbi.nlm.nih.gov/geo/) under accession numbers GSE13092843 and GSE1389644, respectively. Genes related to phagocytosis and specific phenotypes were retrieved from the GeneCards database33 (https://www.genecards.org/) and the MSigDB database37 (https://www.gsea-msigdb.org/gsea/index.jsp).

4- Change the order of Figure numbers. Right now, it starts with Figure-2.

Reply: Thank you very much for your suggestion. We have corrected the numbering of figures and tables in the manuscript.

5- Line 120. Explain what is gene ratio in figure 4?

Reply: GeneRatio represents the ratio of the number of genes included in a certain entry in the enriched gene set to the number of genes in its background database. The numerator is the number of genes enriched in this GO entry, and the denominator is the number of all genes input for enrichment analysis.

6- Figure-3: Mention what software was used to obtain the figures? And mention what is the sample size (n) of normal and COPD in figure 3C?

Reply: Thank you very much for this suggestion, and we have made the following changes in the manuscript:

Figure 2. Construction of weighted gene co-expression network for COPD. A. Soft threshold β = 9 and scale-free topological fit index (R2). B. Dendrogram of sample clustering, with each sample corresponding to a leaf. C. Heat map of module-trait correlation. 18 rows correspond to each of the 18 combined modules, 2 columns correspond to the nor-mal group and the COPD group, and the cells contain the corresponding correlation coefficients and P values. D, E, F. Scatter plots of MM-GS for red, blue, and magenta modules in the COPD group. G, H, I. Scatter plots of MM-GS for red, blue, and magenta modules in the control group. The above graphics were drawn using R software.

The module assignment results are visualized in Figure 2C (24 cases in the normal group and 22 cases in the COPD group).

7- Figure-4: Mention what software was used to obtain the figures?

Reply: Thank you very much for this suggestion, and we have made the following changes in the manuscript:

Figure 3. Identification of key KEGG pathways. A. DEGs mapped to WGCNA key module genes. B. KEGGa, KEGG enrichment results of genes intersecting DEGs with WGCNA key module genes. C. Cross-results of genes predicted by CTD and GeneCards databases. D. KEGGb, KEGG enrichment results of COPD-related genes in disease databases. E. KEGGc, KEGG enrichment results of isoliquiritigenin target genes. The above graphics were drawn using R software.

8- Line 129. What was the source of isoliquiritigenin target genes ? Mention it here.

Reply: Thank you very much for this suggestion, and we have made the following changes in the manuscript:

All genes enriched in the 20 key pathways were combined with the ISO target genes, resulting in 54 predicted therapeutic target genes (Figure 4A). As outlined in the methodology, the predicted molecular targets of ISO were sourced from the Traditional Chinese Medicine System Pharmacology (TCMSP) database, as well as the Comparative Toxicogenomics Database (CTD) and PubChem databases.

9- Line 134. Start the word with a capital letter A.

Reply: Thank you very much for this suggestion, and we have made the following changes in the manuscript:

All genes enriched in the 20 key pathways were combined with the ISO target genes, resulting in 54 predicted therapeutic target genes (Figure 4A).

10- Figure-5. Explain what is the meaning for each of different colors (red, blue, etc) in the legend?

Reply: Thank you very much for this suggestion, and we have made the following changes in the manuscript:

Figure 4. Network construction of predicted therapeutic target genes. A. The intersection of isoliquiritigenin target genes and all genes in the key KEGG pathway. B. The “component-target-pathway” network, in which the blue rectangular nodes are 54 predicted therapeutic target genes, the light red diamond nodes are isoliquiritigenin, and the yellow oval nodes are key KEGG pathways. C. The PPI network constructed by the 54 predicted therapeutic target genes. D. The key network obtained based on MCODE. In figures C and D, a redder node color indicates a greater number of interactions with other genes.

11- Table-1. Mention the source of isoliquiritigenin dataset here. Also, include a column to show the references of the GEO dataset taken for analysis in this paper.

Reply: Thank you very much for this suggestion. Given the numerous isoliquiritigenin genes sourced from various databases, the table has become quite lengthy; therefore, we have included it in the supplementary materials. And we have made the following changes in the manuscript:

Table 3. GEO Dataset Information.

GEO Datasets

Normal non-smoking/case

COPD/case

Species

Method

Organization Source

Type

Platform

GSE13092843

24

22

Homo sapiens

Bronchoalveolar lavage

Alveolar macrophages

Expression profiling by array

GPL570

GSE1389644

24

12

Homo sapiens

Bronchoalveolar lavage

Alveolar macrophages

Expression profiling by array

GPL570

12- Line 353. Revise the figure number and briefly describe summary of workflow chart in the legend of Figure-1.

Reply: Thank you very much for this suggestion, and we have made the following changes in the manuscript:

Figure 9. Graphical workflow of this study. We conducted differential expression analysis and WGCNA on the mRNA expression dataset of COPD patients (GSE130928). COPD-related genes were sourced from the CTD and GeneCards. Isoliquiritigenin-related targets were obtained from the TCMSP, CTD, and PubChem databases. Key pathways were identified from these genes, and those within the key pathways were extracted and intersected to construct a PPI network. Candidate therapeutic targets were identified using the MCODE algorithm and further intersected with genes related to glycolysis and phagocytosis. To identify phenotype-specific therapeutic targets, we applied eight machine learning methods. Clinical evaluations of the key targets included nomogram analysis, ROC analysis, GSEA, immune cell analysis, and molecular docking simulations. The findings were validated in an independent dataset (GSE13896).

13- In the Methods Section, include a paragraph on statistical analysis methods applied of various datasets and how significance was established for each of them.

Reply: Thank you very much for this suggestion, and we have made the following changes in the manuscript:

5.12. statistical analysis

The data were analyzed statistically using R software (version 4.2.1). A significance threshold of P < 0.05 was established, and all tests were two-tailed. Comparisons between the two groups were conducted using a t-test (assuming normality and homogeneity of variance), Welch's t-test (assuming normality but not homogeneity of variance), or the Wilcoxon rank-sum test (non-normally distributed data, nonparametric test). Correlation analyses were performed using Pearson or Spearman correlation coefficients, depending on the data characteristics.

14- Line 460-470. Move this Conclusion section to the last paragraph of Discussion section.

Reply: Thank you for your valuable suggestion; we have relocated the Conclusion section to follow the Discussion section.

15- Need to check thoroughly for spelling and grammatical errors.

Reply: Thank you very much for your suggestion, we have checked it thoroughly for spelling and grammatical errors.

Round 2

Reviewer 1 Report

Comments and Suggestions for Authors

The authors have adequately addressed my comments and concerns, the manuscript can be accepted for publication in present form. 

Reviewer 2 Report

Comments and Suggestions for Authors

The authors have revised the manuscript and satisfactorily addressed the concerns I raised.